# Overview of Non-Alcoholic Fatty Liver Disease (NAFLD) and the Role of Sugary Food Consumption and Other Dietary Components in Its Development

**DOI:** 10.3390/nu13051442

**Published:** 2021-04-24

**Authors:** Pau Vancells Lujan, Esther Viñas Esmel, Emilio Sacanella Meseguer

**Affiliations:** 1Institut d’Investigacions Biomediques August Pi i Sunyer (IDIBAPS), Rosselló 149, 08036 Barcelona, Spain; pauv65@gmail.com (P.V.L.); evinas@clinic.cat (E.V.E.); 2Department of Internal Medicine, Hospital Clínic de Barcelona, Villarroel 170, 08036 Barcelona, Spain

**Keywords:** NAFLD, NASH, nutrition, diabetes, metabolic syndrome, cardiovascular disease, fructose, fatty acids, protein, Mediterranean diet

## Abstract

NAFLD is the world’s most common chronic liver disease, and its increasing prevalence parallels the global rise in diabetes and obesity. It is characterised by fat accumulation in the liver evolving to non-alcoholic steatohepatitis (NASH), an inflammatory subtype that can lead to liver fibrosis and cirrhosis. Currently, there is no effective pharmacotherapeutic treatment for NAFLD. Treatment is therefore based on lifestyle modifications including changes to diet and exercise, although it is unclear what the most effective form of intervention is. The aim of this review, then, is to discuss the role of specific nutrients and the effects of different dietary interventions on NAFLD. It is well established that an unhealthy diet rich in calories, sugars, and saturated fats and low in polyunsaturated fatty acids, fibre, and micronutrients plays a critical role in the development and progression of this disease. However, few clinical trials have evaluated the effects of nutrition interventions on NAFLD. We, therefore, summarise what is currently known about the effects of macronutrients, foods, and dietary patterns on NAFLD prevention and treatment. Most current guidelines recommend low-calorie, plant-based diets, such as the Mediterranean diet, as the most effective dietary pattern to treat NAFLD. More clinical trials are required, however, to identify the best evidence-based dietary treatment approach.

## 1. Introduction

Non-alcoholic fatty liver disease (NAFLD) is a general term used to cover a continuum of liver disorders that are distinguished by evidence of excessive fat in the liver (hepatic steatosis) on imaging or histology (macrovesicular steatosis in >5% of hepatocytes), and the absence of secondary causes (alcohol consumption, medications, hereditary disorders) [1,2]. NAFLD is the most common cause of chronic liver disease worldwide and represents a major, growing, and often overlooked public health problem [3,4]. Fatty liver diseases are closely linked with a globalised economy and an increasingly homogenous sociocultural westernised lifestyle [4,5]. It is also closely related to other metabolic diseases such as type 2 diabetes mellitus (T2DM), obesity, metabolic syndrome (MetS), and dyslipidaemia. In addition, a subtype of NAFLD called non-alcoholic steatohepatitis (NASH) has a potentially progressive course, leading to liver fibrosis, cirrhosis, and/or hepatocellular carcinoma (HCC)—conditions which may require treatment through liver transplantation [6].

Nutrition is the principal contributory factor affecting NAFLD development. Thus, different dietary components could modify its natural course [5,6,7,8,9,10,11,12,13], which means it is important to comprehensively discuss the role of nutrition and its main components in the natural history of NAFLD. Accordingly, the objective of this manuscript is to provide an overview of the general aspects of NAFLD and to review the effects of specific nutrients and dietary patterns in the development of the disease.

## 2. Methods

The objectives of this narrative review are to offer a general overview of NAFLD and to assess the effect of specific nutrients and some dietary patterns on the development of the disease. We searched for scientific studies published in the last 10 years and written in English in PubMed’s website MEDLINE Database, NCBI, and Google Scholar by using specific search terms (‘NAFLD’ or ‘MAFLD’, ‘nutrition’, ‘diabetes’, ‘insulin resistance’, ‘metabolic syndrome’, ‘cardiovascular disease’, ‘fructose’, ‘carbohydrates’, ‘fats’, ‘fibre’, ‘protein’, ‘lipogenesis’, ‘ketogenic diet’, ‘Mediterranean diet’, ‘vegan/vegetarian diet’, ‘DASH’, ‘intermittent fasting’, and ‘microbiota’). In addition, recent reviews, meta-analyses, consensus documents, and guidelines about NAFLD, foods, and its pathophysiological mechanisms were also included. To sensitise the search and select the most pertinent articles, we used the aforementioned keywords and applied different Boolean operators. Eventually, a total of 170 articles were selected, of which 93 were systematic reviews, meta-analyses, consensus documents, or guidelines, and 67 were human observational or intervention studies; a small proportion of animal studies were also included. Human intervention studies with nutritional supplements were eliminated since the aim of the study was to evaluate the effect of real-life natural feeding on the development of NAFLD. The publication, attrition and cointervention bias, and the baseline characteristics of the patients included in the trials should be taken into consideration for the interpretation of the results. 

## 3. Overview of NAFLD

### 3.1. Definition of NAFLD and NASH 

NAFLD is an acquired metabolic disease induced by metabolic stress and characterised by fat deposition in the liver. It advances at different rates of progression among individuals but typically follows a four-stage course. The first stage involves hepatic fat deposition, also known as non-alcoholic fatty liver (NAFL). The second stage, marked by excessive hepatic fat deposition, occurs in approximately 7–30% of NAFLD patients and causes liver inflammation known as NASH. Persistent liver inflammation induces hepatic fibrous tissue formation; this stage is called fibrosis and is characterised by the activation of the hepatic stellate cells and the replacement of hepatocytes with fibrillar collagen and other extracellular matrix proteins which compromise hepatic function and structure. The last stage is cirrhosis, a severe stage of NAFLD during which hepatocytes are completely replaced by fibrosis, leading to liver failure [12,13]. 

The primary histological features of NASH require the presence of steatosis, ballooning, and lobular inflammation in liver biopsy; other histological changes include portal inflammation, polymorphonuclear infiltrates, Mallory–Denk bodies, apoptotic bodies, clear vacuolated nuclei, microvacuolar steatosis, and megamitochondria [14] (Table 1). NASH can be classified from mild (F0–F1) to advanced fibrosis (≥F3, bridging) and finally cirrhosis (F4). Higher degrees of fibrosis influence survival [13] and are related to more progressive disease states that may lead to liver cirrhosis, HCC, and, eventually, a requirement for liver transplantation [6]. In contrast, only a few cases of HCC have been reported in patients with isolated NAFL. 

Both NAFL and NASH remain asymptomatic until late in the disease course; thus, many patients are only identified at advanced stages. This is particularly serious for NASH patients as they have an increased risk of HCC, cardiovascular disease (CVD), and all-cause mortality; hence, early identification of steatohepatitis via accurate diagnostics is imperative in this population. 

Analysis of the metabolic and heterogeneous nature of NAFLD indicates that the nomenclature describing it is inherently problematic. Experts have reached a consensus that NAFLD does not reflect current knowledge, and therefore, metabolic-associated fatty liver disease (MAFLD) has been suggested as a more appropriate overarching term to describe this disease spectrum. However, MAFLD is not yet widely accepted in the literature; hence, in this review, we adhere to the convention of referring to this disease spectrum with the acronym ‘NAFLD’ [15]. 

### 3.2. Epidemiology of NAFLD 

NAFLD is the most prevalent chronic liver disease in the world [4,5]. Several studies have attempted to quantify the true worldwide incidence of NAFLD/NASH, but in view of the extreme differences in study parameters and accurate and accessible testing, a clear and reliable occurrence rate is not currently available [7]. The global prevalence estimations for the general population are 25% for NAFLD and 3–5% for NASH [13,16]. It is important to highlight that NAFLD prevalence has global variations. Furthermore, once they adopt a sedentary lifestyle and overnutrition typical of Western standards, individuals of Asian, Hispanic, Indian, and Native American ancestry seem more susceptible to NAFLD than those of European and African ancestry [4]. Prevalence in the adult population in North America is estimated at 27–34%, 25% in Europe, and 15–20% in Asia. Cases of NAFLD in China, which were estimated to have the greatest relative rate increase to 22.2% in the last decade, reflect the impact of rapid lifestyle changes (westernised diets) due to economic globalisation, such as the adoption of a hypercaloric diet with increased consumption of refined sugars, processed foods (rich in saturated fats), and additives, as well as the lower frequency of daily physical activity. The epidemiological trends of NAFLD parallel the increasing prevalence of obesity, diabetes, and other metabolic disorders [4,5,17]. 

The term ‘lean NAFLD’, meanwhile, refers to the condition of individuals who clinically manifest the disease as a result of having excess visceral adiposity and insulin resistance (IR) rather than as a consequence of making unhealthy lifestyle choices or having a high body mass index (BMI). Across different studies, the reported prevalence of this subset of NAFLD ranges from 7–20% of NAFLD subjects. It was initially described in Asian populations, although it has since been recognised globally [4] (Table 1).

Although NAFLD mainly affects the middle aged and elderly, it is noteworthy that it can affect people of all ages. In fact, the prevalence of NAFLD in young populations is quickly increasing worldwide due to sedentarism and unhealthy dietary patterns [4,18]. Moreover, the influence of sex on NAFLD prevalence is difficult to determine, but overall, the data indicate that male sex and menopausal status are associated with a higher risk of NAFLD independent of age and metabolic factors [19]. As mentioned before, its incidence is growing worldwide in parallel with overweight, obesity, T2DM, and MetS [5,13,16]. NAFLD has a bidirectional association with MetS, and it is frequently recognised as the hepatic manifestation of MetS [20].

In addition, NAFLD is associated with an increased risk of liver-related, cardiovascular, and all-cause mortality [21]. Patients with NAFLD also have an increased risk for T2DM, atherosclerosis, CVD, and chronic kidney disease [5,13,16,22]. Consequently, they have a shorter lifespan and may experience a lower quality of life. Due to both its frequency and potential severity, NAFLD is considered a health problem of the first magnitude [3,5].

### 3.3. Pathogenesis of NAFLD

The pathogenesis of NAFLD is not completely understood. Initially, a classical ‘two-hit theory’ was proposed to explain its development; however, this hypothesis is now considered obsolete [2,23]. Today a ‘multiple-hit model’ is widely accepted; this hypothesis suggests that multiple factors act together on genetically predisposed subjects to induce NAFLD [23,24]. These hits include IR, elevated plasma-free fatty acids, oxidative stress, liver inflammation, adipose tissue-secreted hormones, nutritional factors, gut microbiota, and genetic and epigenetic factors. Among the proposed hit factors, many of them can interact with each other, although liver fat accumulation caused by obesity and IR appears to represent the first hits. The role of microbiota in NAFLD pathogenesis is described in the next section.

The multiple-hit theory begins with fat accumulating in hepatocytes as cytoplasmic lipid droplets, initiating simple steatosis. Liver steatosis is reversible and occurs mainly via four mechanisms: increased fatty acid uptake to the liver; reduced-fat transport in the form of very low-density lipoprotein (VLDL) triglycerides; decreased free fatty acid β-oxidation; and increased de novo lipogenesis (DNL) in the liver (Figure 1) [25]. 

IR plays a central role by causing massive metabolic dysregulation of NAFLD, which initiates and aggravates hepatic steatosis [26]. IR prevents adequate suppression of hepatic glucose production in the presence of preserved lipogenesis stimulation, whereas insulin cannot deactivate lipolysis in the adipose tissue [27]. 

Another characteristic of patients with NAFLD is that their inflammatory pathways are more activated [28]; this activation of the Kupffer cells occurs via two main classical inflammatory pathways: JNK-AP-1 and IKK-NF-κBDl. The increased levels of free fatty acids cause lipotoxicity and IR and, together with pathogenic drivers such as endotoxins and xenobiotics, activate the release of proinflammatory cytokines systemically and locally in the liver [29,30]. Inflammation is also triggered by endotoxins present in patients with dysbiosis via the portal vein. Furthermore, lipotoxicity of accumulated lipids and innate immune system activation are major drivers of NASH. Lipid-induced stress activates inflammatory processes (including the release of proinflammatory extracellular vesicles and cell death) and immune mechanisms involving macrophages, dendritic cells, and lymphocytes; these act as central drivers of inflammation that recognise damage- and pathogen-associated molecular patterns and contribute to the progression of the inflammatory cascade [31]. Recently, the roles of the adipose tissue–liver axis and gut–liver axis have been highlighted as critical drivers of inflammation and fibrosis in NAFLD [24]. 

Additionally, genetic variations likely play a role in NAFLD. It has been found that an allele in gene variant rs738409 of PNPLA3 (patatin-like phospholipase domain-containing protein 3) contributes to ancestry-related and interindividual differences in hepatic fat content and predisposition to NAFLD; it also enhances NAFLD severity across the entire histological spectrum, leading to cirrhosis in a higher proportion of subjects. Other gene polymorphisms show a relationship between fructose and NAFLD, such as the transmembrane glucokinase regulatory gene (GCKR) which regulates the glycolytic pathway, and the 6-superfamily member 2 (TM6SF2) which regulates VLDL secretion. More genetic studies are needed to find clear causation in this area despite early data suggesting that genetic polymorphisms interacting with fructose play a role in NAFLD pathogenesis [32]. 

### 3.4. Gut Microbiota and NAFLD 

Gut microbiota (GM), which includes bacteria, viruses, and fungi coexisting in the gut, has crucial physiological functions in host digestion, immunity, and metabolism. It, therefore, contributes to metabolism and plays an important role in obtaining energy and nutrients from the diet [33]. GM is highly heterogeneous and greatly influenced by the diet and lifestyle of the host. In fact, westernised diets (which are rich in sugary food and fats) negatively influence GM and can cause dysbiosis, leading to intestinal inflammation, collapse of the intestinal barrier, and translocation of microbial products [34]. Some species of *Bifidobacterium* spp., *Akkermansia* spp., and *Lactobacillus* spp. are beneficial, whereas *Bacteroides* spp. and *Ruminococcus* spp. are implicated in negative health outcomes [35]. Core microbial diversity and the ratio of *Firmicutes* spp. to *Bacteroidetes* spp. are general indicators of health.

Various studies in both animals and humans have linked alterations in the microbiota with NAFLD development [36]. In fact, direct faecal microbiota transplantation (FMT) from obese mice with or without steatosis to germ-free recipients replicated some NAFLD alterations [37]. Moreover, FMT from obese women with NAFLD to conventional mice fed a chow diet resulted in increased IHTGs within 2 weeks [38]. It is noteworthy that preliminary FMT studies from human allogeneic vegan donors to NAFLD patients showed inverse results, exhibiting improved necroinflammatory histology and changes in hepatic gene expression involved in inflammation and lipid metabolism [39]. 

Some authors suggest that GM alterations induce greater intestinal permeability and LPS release in the gut, which could activate the immune system, whereas microbially produced metabolites (short-chain fatty acids, endotoxins, trimethylamine-N-oxide, choline, or ethanol) could regulate bile acid metabolism [40]. Bacterial endotoxins can go through the portal vein to the liver, inducing inflammation and IR and promoting the development of fatty liver [41]. This process, known as endotoxemia, is present in many NAFLD patients. In fact, NAFLD can improve with short-term antibiotic therapy by eliminating harmful microbiota, while long-term therapy can result in bacterial resistance, reducing the efficacy of the drug and increasing the risk of secondary infections [42,43].

A recent systematic review and meta-analysis found an interaction between GM, related metabolites, and inflammatory factors with NAFLD [36]. However, no clear microbiome signature has yet been discovered in NAFLD patients [35,44]. The only consistent observations from the data are that greater amounts of Proteobacteria (especially Escherichia coli and other Enterobacteriaceae spp) and lower quantity of Bacteroidetes are observed in patients suffering from NAFLD, whereas a decreased Firmicutes:Bacteroidetes ratio is also noted [44,45]. While animal studies demonstrate a potential causal role, human studies are only just starting to describe microbiome signatures in NAFLD. GM signature inconsistencies in NAFLD patients may be explained by ethnicity, different GM assessment methodologies, population characteristics, drug consumption, and circadian rhythm. Furthermore, microbial signatures are found to be different depending on the stage of NAFLD; for example, Bacteroides vulgatus and Escherichia coli have a greater expression in advanced fibrosis (F3–F4) [46]. 

As mentioned above, diet can change the microbiome. Thus, HFruDs or high-fat diets are capable of inducing dysbiosis (marked by decreased microbial diversity and increased Firmicutes:Bacteroidetes ratio) in animals, which is accompanied by local and systemic inflammation and liver fat infiltration [34,36,47]. By contrast, rodents fed a high-fibre diet showed a reduction in hepatic inflammation and liver fat [48]. A potential mechanism of a high-saturated-fat diet triggering NAFLD via GM is the enhancement of energy extraction from the diet; this occurs through upregulation of whole-body metabolism to fatty acid uptake from adipose tissue and switching fatty acid metabolism from oxidation to de novo synthesis [35]. 

Consequently, modulating GM to reverse dysbiosis could be a potential therapeutic approach for treating NAFLD. Supplementation with probiotics, prebiotics, or synbiotics has been proven to treat dysbiosis and improve NAFLD [49]. Studies performed in animals suggested that probiotics could improve hepatic steatosis, fibrosis, and some metabolic markers [50,51]. Preliminary data in human studies confirmed these findings, although well-designed randomised controlled trials are needed to define the efficacy of these products in human beings for NAFLD treatment further [52]. 

In summary, GM composition and function are strongly intertwined with the pathogenesis and the progression of NAFLD. Restoring microbiota is recognised as a strategy for NAFLD treatment. However, most of the evidence related to this remains experimentally based on murine models and has several limitations. Therefore, precisely defined large-scale human studies that consider confounding factors and that reduce sample heterogeneity are needed to identify the exact mechanisms and recognise the best probiotic strains to treat NAFLD.

### 3.5. Diagnosis of NAFLD

The 2016 NAFLD management guidelines from the European Association for the Study of the Liver, the European Association for the Study of Diabetes, and the European Association for the Study of Obesity (EASL–EASD–EASO) propose an initial evaluation of patients in whom NAFLD is suspected through medical interview, physical exam, blood tests (mainly liver enzymes, blood glucose, glycated haemoglobin, lipid profile, and others if indicated) and abdominal ultrasonography [14].

The incidental discovery of steatosis should lead to evaluation of the family and personal history of NAFLD-associated diseases and the exclusion of secondary causes. Similarly, the presence of obesity/T2DM or incidental findings of elevated liver enzymes in patients with metabolic risk factors should prompt non-invasive screening to predict steatosis, NASH, and fibrosis [14]. For the diagnosis of NAFLD, clinicians should not rely solely on liver enzyme levels because they have low sensitivity (50%) and specificity (61%) to predict NASH [53,54].

Ultrasonography (US) has shown good sensitivity (85%) and specificity (94%) in detecting moderate and severe steatosis and is the first-line, non-invasive modality for diagnosing NAFLD. Nevertheless, US detects less than 20% of mild hepatic steatosis (at least 20% of hepatocytes have been fatty transformations), while magnetic resonance imaging (MRI) can detect as little as 5% steatosis [14,53]. 

Transient elastography (TE, or FibroScan^®^) is an ultrasound-based modality with high sensitivity and specificity that measures liver stiffness as a surrogate marker for hepatic fibrosis [55]. Other non-invasive methods are controlled attenuation parameter (CAP), point quantification shear wave elastography (Pswe), magnetic resonance elastography (MRE), and shear wave elastography (SWE) [55,56,57,58]. Moreover, advanced MRI can measure the proton density fat fraction (PDFF), which is a quantitative indicator of hepatic fat content highly correlated with biopsy-proven steatosis (96% sensitivity and 100% specificity) [53,54,55,56].

Several non-invasive scoring systems have been developed to optimise the early identification of high-risk individuals [53]. Those with the best-validated scores include the fatty liver index (FLI) and the NAFLD liver fat score (NLFS), which variably predict metabolic, hepatic, and cardiovascular outcomes, as well as the presence of steatosis [14] (these scores are described in Table 2). The FLI was developed and validated to select elderly subjects for ultrasonography to detect NAFLD. The recommended cut-offs are as follows: FLI < 30 to rule out (87% sensitivity) and FLI ≥ 60 to rule in hepatic steatosis (specificity 86%) [38,39]. The NLFS evaluates liver fat content measurements and has shown satisfactory accuracy in diagnosing NAFLD, (95% sensitivity), although in the general population these tools offer lower diagnostic efficacy (70–80%) [55].

Other fibrosis biomarkers and scores include the NAFLD fibrosis score (NFS) and the fibrosis-4 (FIB-4) index, which are acceptable non-invasive procedures for identifying cases at low risk of advanced fibrosis/cirrhosis and may predict overall mortality, cardiovascular mortality, and liver-related mortality [14,53].

Multiple circulating biomarkers have also been used for diagnosis, such as caspase-generated cytokeratin-18 (CK-18) fragment concentration, which is a biomarker of hepatocyte apoptosis in the context of NASH [54,59]. However, neither serological nor radiological biomarkers are sufficiently accurate to distinguish NASH from NAFLD or to reliably detect early stages of fibrosis [53]. Currently, histological analysis of a liver biopsy sample is recognised as the gold standard to differentiate NAFLD from NASH [14,54,55,56]. 

After NAFLD has been confirmed, patients can be stratified by their progression risk to determine the course of treatment [54]. Finally, monitoring hepatic disease and its underlying metabolic conditions should include routine biochemistry, assessment of comorbidities, and non-invasive monitoring of fibrosis. NAFLD patients without worsening metabolic risk factors should be monitored at 2–3-year intervals [14]. 

### 3.6. NAFLD and Comorbidities

#### 3.6.1. NAFLD and Metabolic Comorbidities

There is a high burden of metabolic comorbidities associated with NAFLD. Firstly, NAFLD is tightly associated with IR and MetS. Secondly, BMI and waist circumference, a measure of visceral adiposity, have been positively related to NAFLD and may predict advanced disease, particularly in the elderly. Thirdly, common comorbidities of obesity (such as T2DM) and sleep apnoea, polycystic ovary syndrome, and other endocrine disorders (such as hypogonadism) further drive NAFLD prevalence and severity, progression to NASH, advanced fibrosis, and HCC. Therefore, screening for diabetes in patients with NAFLD is mandatory, and the reverse—screening for NAFLD in patients with diabetes—is also mandatory [14]. 

A large meta-analysis concluded that NAFLD was significantly associated with an almost twofold increased risk of incident T2DM and MetS over a median follow-up of 5 years, suggesting that NAFLD may also be a risk factor for subsequent development of T2DM and MetS [60]. Another meta-analysis demonstrated the variability in the distribution of different comorbidities between NAFLD and NASH (Table 3). 

#### 3.6.2. NAFLD and Cardiovascular Disease

The prevalence and incidence rate of CVD is higher in patients with NAFLD, compared to the general population; the risk increases further in NASH and advanced fibrosis. Therefore, CVD should be ruled out in individuals with NAFLD regardless of the presence of traditional risk factors. Conversely, NAFLD screening should be performed in persons at high risk for CVD [14]. Individuals with NAFLD have a 1.5–6 times higher risk of CVD and CVD-related mortality and diabetes, mostly independent of other known risk factors [53]. Additionally, a recent meta-analysis found evidence that diabetic patients with NAFLD had more than twofold higher risk for CVD, compared to patients without NAFLD (odds ratio [OR] 2.20, 95% confidence interval [CI] 1.67–2.90), suggesting a synergistic effect between NAFLD and T2DM on the risk of CVD. However, it remains uncertain whether NAFLD is associated with CVD as a result of coexisting cardiovascular factors, or if it independently raises CVD risk as a proatherogenic stimulus [61].

#### 3.6.3. NAFLD and Other Comorbidities

HCC has also been reported in NAFLD and, most prevalently, in NASH or cryptogenic cirrhosis. However, a recent meta-analysis showed that the annual incidence of HCC in NAFLD patients was 0.44 per 1000 person years, suggesting that HCC is a rare complication in NAFLD [13]. On the other hand, Helicobacter pylori infection is a more common factor contributing to IR promotion and NAFLD progression via upregulating the expression of various inflammatory factors, such as tumour necrosis factor, C-reactive protein, and interleukin. Moreover, Helicobacter pylori can retrograde into the liver through the hepatic bile duct, leading to chronic liver inflammation that causes hepatic cell damage and necrosis [62]. 

### 3.7. Prognosis

NAFLD has a nonlinear, slowly progressive course and is probably more dynamic than previously thought. The fibrosis progression rate is estimated to consist of one stage of progression over 14 years for patients with NAFLD and 7 years for patients with NASH [14,16]. In addition, NAFLD and NASH progress to cirrhosis in 2–3% and 15–20% of patients, respectively, over a 10- to 20-year time frame [54].

Several studies have demonstrated that NAFLD patients have significantly increased mortality, compared with the general population (hazard ratio [HR] 1.29, CI 1.04–1.59), with increased risk of CVD (HR 1.55, CI 1.11–2.15), HCC (HR 6.55, CI 2.14–20.03), and cirrhosis (HR 3.2, CI 1.05–9.81) [63]. 

CVD remains the most common cause of death in NAFLD patients, with the worst prognosis being found in patients with stage 3 or 4 fibrosis at baseline [63]. On the other hand, patients with NAFLD have a higher risk of death from liver disease, compared to those without, and the risk grows exponentially as the fibrosis stage progresses [14]. The fibrosis stage is the most important prognostic factor in NAFLD and independently predicts increased liver-related outcomes and mortality [14,53,63]. In fact, rates of liver-specific morbidity and overall mortality do not generally increase among patients with stage 0–2 fibrosis, whereas increased mortality is strongly correlated with stage 3–4 fibrosis (HR 3.3, CI 2.27–4.76) [53,63]. The example of the USA demonstrates the growing seriousness of NAFLD: there it is the second-leading indication for HCC-related transplantation and the third most common cause for liver transplantation (and on track to become the most common cause) [14,63].

## 4. Nutrition and NAFLD 

Poor nutrition is the leading contributing factor to the development of NAFLD. This means that a healthy diet, combined with weight loss and increased physical activity, forms the most effective therapeutic approach for NAFLD management. In fact, since no standardised or specific medications have yet been approved to treat NAFLD, treatment currently centres around improving patients’ diets and exercise habits.

Dietary intervention (especially the Mediterranean diet) is effective in normalising aminotransferases and lowering intrahepatic fat, whilst exercise improves insulin sensitivity and decreases BMI [64,65]. NAFLD patients should be advised to maintain a low-calorie diet and stop alcohol and tobacco consumption [66]. A daily caloric restriction of 500–1000 kcal is an extremely effective intervention as primary or secondary prevention against NAFLD. A weight-loss reduction of 3% to 5%, meanwhile, is also associated with decreased NAFLD, but a greater reduction in weight (7–10%) is necessary to achieve NAFLD remission and fibrosis regression. The goal of calorie restriction should therefore be to achieve ≥10% overall body weight loss [67], though some authors advise to not exceed 1.6 kg/week weight reduction so as to avoid a worsening of fibrosis and hepatocyte necrosis [68].

The overall goal is to find the best dietary pattern and macronutrient composition to prevent, attenuate, or reverse hepatic steatosis and its progression to steatohepatitis. Diets that can improve IR, oxidative stress, or inflammation are potentially good candidates to treat NAFLD. Below, we summarise what is currently known about the relationship between the core nutritional elements (carbohydrates, fats, and proteins), food groups, and dietary patterns, and the likelihood of NAFLD.

### 4.1. Carbohydrate Intake and NAFLD 

Carbohydrates are classified as simple (fructose, glucose, galactose) and complex (starch). There is evidence suggesting that lower carbohydrate intake (≤40% of daily energy intake) may be beneficial for NAFLD patients [69] and that it is preferable to avoid consumption of carbohydrates with a high glycaemic index [70]. However, most of the conducted studies focus on fructose consumption and its relationship with NAFLD due to its unique hepatic metabolism and because the rise in the consumption of added sugars, particularly fructose, parallels the increasing incidence of NAFLD (Figure 2). 

Fructose is present in natural foods (fruits, vegetables, honey) and in processed foods (juices, nectars, other beverages) [71]. In addition, fructose is a primary component in the most widely used sweeteners (sucrose or high fructose corn syrup [HFCS]). Fructose consumption has increased by 30% in the last 40 years and by 500% over the last century due to the increased consumption of processed foods [72]. Sugar-sweetened beverage (SSB) intake increased by more than 40% from 1990 to 2016 [73]. Today, added sugar intake makes up 15% of total daily calories in the average Western diet [17]. In parallel, there has been a progressively higher incidence and prevalence of obesity, NAFLD, T2DM, and MetS. The increased consumption of added sugars, particularly fructose, is a major underlying cause of chronic metabolic diseases, including NAFLD, T2DM, obesity, hypertension, and CVD [74,75,76,77]. Numerous experimental and clinical studies have found that high fructose consumption is a major risk factor for NAFLD and its consequences [78,79]. 

Observational studies clearly reveal a close relationship between overconsumption of added sugars and the development of NAFLD in adults and children. A systematic review and meta-analysis of seven studies (six cross-sectional studies and one cohort study) involving 4639 subjects demonstrated that SSB consumers had a 53% increased risk of developing NAFLD compared with non-consumers [79]. Another systematic review and meta-analysis of 12 studies involving 35,705 participants showed that higher consumption of SSBs was positively associated with a 40% increase of NAFLD [11]. Notably, this study found that consumption of SSBs has a dose-dependent effect on the risk of NAFLD. Specifically, low doses (<1 cup/week), medium doses (1–6 cups/week), and high doses (≥7 cups/week) of SSBs significantly increased the relative risk of NAFLD by 14%, 26%, and 53%, respectively. This result was mainly drawn from a cross-sectional study involving 26,790 Chinese adults and showed how a small amount of soft-drink intake is associated with a 14–16% increase in the prevalence of NAFLD, even when adjusted for the presence of MetS [80]. The main limitation of these studies was that they only reflected NAFLD trends in China since almost all data was obtained from the Chinese population. However, other studies with different populations, such as the Framingham Heart Study (*n* = 2600), also showed a dose–response association [81]. 

Lastly, a more recent systematic review and meta-analysis (*n* = 9887) that analysed the influence of all foods on NAFLD development found that dietary intake of added fructose specifically (in the form of sucrose or HFCS) was positively correlated with the likelihood of NAFLD (OR 1.29, CI 95% 0.19–1.40) [82]. It is not yet known if there is a safety threshold for SSB or sugary food consumption for NAFLD prevention [83]. 

Interventional trials also suggest a role for fructose in NAFLD. For instance, daily consumption of SSBs for 6 months by overweight subjects (*n* = 47) resulted in the accumulation of liver fat as proven by magnetic resonance spectroscopy [84]. Even short-term interventions demonstrated that consumption of a high-fructose diet (HFruD) had a significant lipogenic effect on the liver [85]. This role was confirmed by a small study that found (*n* = 16) administering a high-sugar-hypercaloric diet (>1000 kcal simple carbohydrates/day) to subjects for 3 weeks induced DNL, as measured by the lipogenic index [85]. 

Conversely, a 9-day period of isocaloric fructose restriction in obese children (*n* = 41) with habitual high fructose consumption showed reductions in liver fat and DNL when compared to controls fed an isocaloric diet [86]. Furthermore, in a subgroup of the same sample, there was an improvement in metabolic parameters such as diastolic blood pressure, serum triglycerides, and IR [87]. While there is consistent evidence for fructose’s causative role in liver fat deposition in a hypercaloric setting, there is conflicting evidence about the impact of carbohydrates on NAFLD in the context of hypocaloric or isocaloric diets. A systematic review and meta-analysis of six observational studies and 21 intervention studies concluded that there was insufficiently robust evidence available to demonstrate that high intake of fructose, HFCS, or sucrose is associated with a higher incidence of NAFLD and suggested that the relationship between fructose intake and NAFLD could be confounded by excessive energy intake [88]. In accordance with this, a recent short-term interventional study found that a high-fructose diet (150 g a day for 8 weeks) in an isocaloric context does not have negative health impacts on weight, liver health, cholesterol, insulin sensitivity, or glucose/blood sugar tolerance in healthy individuals [89]. In this trial with healthy individuals, no change in intramuscular or intrahepatic fat was observed, but there were higher levels of serum triglycerides and DNL. Additionally, a randomised trial that compared a hypocaloric low-carbohydrate diet with a classical hypocaloric low-fat diet showed comparable beneficial results from both interventions; this suggests that total energy deficit is the mediating factor for decreasing liver fat and that carbohydrate consumption has little impact [90].

However, these studies have some limitations [88,89,91]. The follow-up period in these trials is short (mostly 4 weeks or less), whereas we know from animal models that fatty liver develops after at least 8–24 weeks on a high-fructose diet [92]. Other important limitations are the small sample size, poor quality, heterogeneity of the studies, and the fact that most participants were healthy men, which limits the ability to apply the implications of these findings to other populations at high risk for developing NAFLD (T2DM and obese patients). 

It is noteworthy that one cross-sectional study from Finland found an inverse relationship between fructose intake and NAFLD. This is because the population studied obtained fructose mostly through fruits and not through sugary drinks [93]. Although fruits contain considerable amounts of simple sugars such as fructose, they are unlikely to induce NAFLD and related diseases for various reasons [94]. Fruits have a lower fructose content per gram (compared to soft drinks) and contain beneficial phytochemicals, micronutrients, and fibre (which is a contributor to overall metabolic health). 

In summary, the scientific evidence available to date suggests that chronic, excessive fructose intake is likely a major contributor to NAFLD pathogenesis, especially in genetically predisposed subjects and in the context of hypercaloric diets. Therefore, doctors should encourage their patients with (or at high risk for) NAFLD to reduce added sugar intake, especially fructose.

### 4.2. Fat Intake and NAFLD

Dietary intervention studies performed in animal and human models have analysed the effect of fats on NAFLD [100]. Most studies suggest that high fat consumption plays a role in NAFLD’s pathogenesis [101,102]. However, monounsaturated, polyunsaturated, and saturated fatty acids (MUFAs, PUFAs, SFAs, respectively) or trans fatty acids (TFAs) do not exert the same effect on the liver, and the main source of each of them is different. 

Interventional studies performed in mice show that a long-term high-fat diet promotes NAFLD development [100,101,102,103,104]. For example, an 80-week high-fat diet (60% fat, 20% protein, and 20% carbohydrate) in mice led to obesity and IR, while histological analysis demonstrated liver steatosis, cell injury, inflammation, and fibrosis [103]. 

Epidemiological studies show that NAFLD patients typically have greater intakes of SFAs and cholesterol and a lower PUFA intake, compared to matched controls without NAFLD [105,106]. The findings of some experimental studies with human beings confirm these results. In an 8-week double-blind randomised trial with obese individuals (*n* = 61), SFAs increased liver fat content (50% relative increase) and serum ceramides, whereas PUFAs did not [107]. Additionally, a 10-week randomised trial (*n* = 67) comparing two isocaloric diets containing Omega-6 PUFAs or SFAs showed that Omega-6 PUFAs supplementation caused a reduction in hepatic steatosis, insulin levels, and inflammatory markers when compared with SFAs, though no changes in weight were observed [108]. In another small clinical trial, 38 overweight subjects were overfed for 3 weeks with different macronutrients (unsaturated fats, SFAs, and simple carbohydrates) and showed intrahepatic triglyceride (IHTG) increases of 55%, 33%, and 15% in the SFA, carbohydrate, and unsaturated fat groups, respectively. Furthermore, SFAs induced IR and endotoxemia and significantly increased multiple plasma ceramides [109]. This effect was also confirmed in healthy adults taking a palm oil bolus, which resulted in a measurable increase in IHTG, energy metabolism, and IR [110]. 

In general, NAFLD patients consume less Omega-3 PUFAs and a higher Omega-6:Omega-3 ratio, compared to healthy controls [9]. Dietary Omega-3 PUFAs are mainly found in fish, seeds, walnuts, and some plant oils. A cross-sectional study in the paediatric population (*n* = 223) found that most children with NAFLD have insufficient intake of Omega-3 PUFAs (<200 mg/day) [111]. Indeed, as precursors of eicosanoids, Omega-3 fatty acids have an anti-inflammatory effect, regulate hepatic lipid composition, and improve IR [112,113], whereas Omega-6 fatty acids are proinflammatory and could have a negative effect on NAFLD [114]. 

In fact, many systematic reviews and meta-analyses of randomised controlled trials have addressed the topic of Omega-3 supplementation and its effect on patients suffering from NAFLD. These studies conclude that Omega 3 PUFAs supplementation (>3 g/day) is useful for the reduction of liver fat, hepatic enzymes, BMI, triglycerides, and cholesterol [115,116]. This potentially sets Omega-3 food supplementation as a safe, viable, and effective intervention to help treat NAFLD. 

Regarding the role of MUFA intake in the prevention of NAFLD, there are heterogeneous results: observational studies suggest a neutral effect [117,118], while interventional studies such as the PREDIMED study show a beneficial effect with extra virgin olive oil (EVOO) supplementation (90% of fat in EVOO is MUFA) [119]. Some small, short-term randomised clinical trials have also shown clear positive results [120,121]. Thus, the consumption of a high-MUFA diet (28% of total caloric intake) for 8 weeks by T2DM patients reduced liver fat by 29% without body weight changes, in comparison to a baseline diet moderately rich in SFAs (13% of total energy). Recently, a double-blind randomised controlled clinical trial with NAFLD patients (*n* = 66) found that consumption of 20 g/day of olive oil attenuated fatty liver grade and reduced body fat percentage [122]. It has been reported that MUFA may prevent the development of NAFLD by improving plasma lipid levels, reducing body fat accumulation, and decreasing postprandial adiponectin expression [123]. Based on these results, the EASL-EAS-EASO Clinical Guidelines recommend the Mediterranean diet for NAFLD subjects due to its high content of MUFAs [14].

Finally, TFA consumption has a pro-oxidative effect and is associated with an increased risk for CVD, IR, obesity, and systemic inflammation and may also damage the liver [124]. These data are mainly based on population studies such as the Rotterdam study (*n* = 3882), which indicates that TFAs found in desserts and processed foods were associated with a higher prevalence of NAFLD [118]. Due to its unhealthy effects, there is no human clinical trial to assess the effect of TFA intake in the liver. 

In summary, the effect of fat intake on NAFLD development depends on the type of fat. Some fats (MUFA and PUFA) protect against NAFLD whereas others (SFA and TFA) have a negative effect on NAFLD. However, most of the studies are observational, and only a limited number of small clinical trials have been published to date. Although the evidence is of low-moderate quality, and there are still many questions to be resolved, clinicians should nevertheless advise their patients to lower SFA intake, eliminate TFAs, and increase Omega-3 PUFA to reduce the incidence of NAFLD.

### 4.3. Protein Intake and NAFLD

Little is known about the influence of protein and amino acids on NAFLD since most of the intervention and mechanistic studies have focused on carbohydrates and fats. Furthermore, intervention and observational studies on the relationship between protein and NAFLD show conflicting results. Some studies found evidence that higher protein intakes might have a negative effect on NAFLD, while other studies showed a neutral or positive effect [125,126]. These inconsistencies might be explained by the type of protein consumed, as animal protein seems to have a negative effect on NAFLD, while plant protein has an inverse association [125]. It is well established that high meat intake, especially of red meats and processed meats, is associated with IR, T2DM, and CVD [127,128,129]. NAFLD has also been associated with red and processed meat consumption even when intakes are low [130,131]. However, this effect may be because of the saturated fat content, salt, additives, and cooking methods, more than the effect of the protein itself. 

In conclusion, the effects of protein on the development of NAFLD remain unclear, but they seem positive, especially with vegetable origin proteins, which is likely due to their high fibre and phytonutrient content. The specific problems associated with animal proteins may be related to the fact that they are usually consumed jointly with SFAs (meat products). 

### 4.4. Fibre Intake and NAFLD

The relationship between fibre intake and NAFLD has been assessed in several observational studies and in a small number of small clinical trials with conflicting results. Although most of the studies suggest that high daily fibre consumption is associated with a preventive effect against NAFLD, other authors have not confirmed these results [132]. Recently, a study performed in a Dutch population found that subjects with a high fatty liver index (FLI) consumed less fibre daily, compared to those with a low FLI [117]. Similar results were also obtained in a subgroup of 6613 US adults participating in the National Health and Nutrition Examination Survey [132]. However, it is not fully known whether the protective effect of fibre against NAFLD is due to an indirect action through modulation of the microbiome or if it is a result of the direct anti-inflammatory properties of fibre [71].

### 4.5. Food Groups

Several epidemiological studies have attempted to assess the relationship between different food groups and the risk of NAFLD; however, the results obtained were quite heterogeneous. Recently, Chinese investigators performed a meta-analysis to study the association between eleven food groups (red meat, soft drinks, nuts, whole grains, refined grains, fish, fruits, vegetables, eggs, dairy products, and legumes) and the probability of developing NAFLD. They collected 24 studies (15 cross-sectional and 9 case–control studies) and made the following conclusions: (1) there is a positive association between red meat and soft drink consumption and the risk of NAFLD, (2) there is a negative association between nut consumption and the likelihood of NAFLD, and (3) no significant causal relationship was observed between the other food groups and NAFLD. Although this study had some limitations (reduced number of studies, risk of bias could not be tested, most studies did not stratify food intake), the results obtained were consistent with the current recommended guidelines for treating NAFLD [14,82].

### 4.6. Dietary Patterns and NAFLD 

#### 4.6.1. Low and Very Low Carbohydrate Diets (Ketogenic Diet)

Low/very low carbohydrate diets are dietary patterns that restrict the overall intake of carbohydrates. In the ketogenic diet (KD), carbohydrate content is kept below 10% of total daily caloric intake (20–50 g of carbohydrate/day) [133], although the specific macronutrient composition may vary. The KD is a powerful tool to induce weight loss and achieve greater long-term weight reduction, compared to low-fat diets [134]. 

With respect to NAFLD, the effect of low-carbohydrate diets remains extremely controversial [68]. Several small clinical trials have confirmed that obese patients following a KD improve liver steatosis, inflammation, IR, and dyslipidaemia and achieve a significant reduction in body weight in the short term [135,136]. Nevertheless, some authors suggest that these effects are lost in the long term (>12 months) [137]. However, the results obtained with a KD are quite heterogeneous, and the samples are very small; thus, it is difficult to draw conclusions that can be applied to the general population. In addition, it is not possible to determine whether the beneficial effects are due to weight loss, calorie restriction, or changes in macronutrient distribution. Some authors suggest that the KD could be a practical short-term approach to NAFLD because it may offer an opportunity to exclude sugary foods from the diet and because the greater weight reduction associated with a KD makes it easier to achieve a normal BMI [138].

#### 4.6.2. Mediterranean Diet 

The Mediterranean diet (MedDiet) is composed in the following way: it is typically rich in vegetables, fruits, whole grains, legumes, nuts, and seeds, and EVOO; it is moderate in fish and other meats, dairy products, and red wine; and it involves a low intake of eggs, red meat, and sweets. As a result, it is a diet rich in antioxidants and fibre, with little saturated fat or animal protein, and with an appropriate Omega-3:Omega-6 fatty acid ratio [139]. It has several health benefits, including a lower incidence of CVD and components of MetS [69,140,141]. In addition, several studies (observational studies and short-term trials) have demonstrated that the MedDiet is beneficial due to its effect on most risk factors for NAFLD (body weight, serum triglycerides, and IR) [142,143]. A recent short-term, small clinical trial (*n* = 49) showed that an ad libitum low-fat diet and MedDiet reduced hepatic steatosis by approximately 25% and was associated with minimal weight loss (−2.1 kg) [144]. Another clinical trial with 63 overweight or obese participants observed that the combination of a MedDiet and increased physical activity had a greater effect against NAFLD than a MedDiet alone [145]. 

Some authors suggest that the use of the Mediterranean adequacy index—which is obtained from the ratio of the combined percentage of total energy from Mediterranean foods and the total energy from non-Mediterranean foods—is a useful tool to assess adherence to the MedDiet. A daily score greater than 5 is necessary to obtain a health benefit [96]. Today, a growing body of evidence suggests that the MedDiet, in combination with exercise and behaviour therapy, may be the reference nutritional pattern to prevent and treat NAFLD [143]. In fact, most guidelines recommend the MedDiet as the best dietary pattern against liver steatosis [146,147]. However, it is necessary to perform more clinical trials with larger sample sizes in order to obtain the best scientific evidence on this issue, standardise the MedDiet characteristics, and develop specific tools to assess adherence to this type of diet. 

#### 4.6.3. Vegetarian and Vegan Diets 

These diets are based on low or no consumption of animal products and a high intake of vegetables, legumes, fruits, whole grains, and fibre, which are all rich in polyphenols with antioxidant and anti-inflammatory properties. Many studies have noted that plant-based diets have a protective effect against multiple diseases related to NAFLD such as T2DM, MetS, hypertension, obesity, CVD, and all-cause mortality [148,149,150,151]. Furthermore, several randomised controlled studies demonstrated that a vegan diet could be more efficient for weight loss than other eating patterns including a vegetarian diet or a MedDiet [152,153]. Population studies such as NHANES and the Rotterdam study indicated that high adherence to a plant-based diet was associated with improvement in risk factors (IR, BMI) related to NAFLD development [154,155]. A cross-sectional study (*n*=3279) analysed food substitution in vegetarians and found that meat eaters had a 12% greater risk of developing NAFLD than vegetarians, suggesting that replacing animal protein with plant-based protein may prevent fatty liver [156]. Although these dietary patterns involve higher fructose intake, they do not seem to be associated with major incidences of NAFLD. This paradoxical situation may be due to the fact that the main source of fructose is whole fruit [157]. In summary, there is a lack of randomised controlled clinical trials assessing the effect of a vegan/vegetarian diet against NAFLD. Although the evidence is of low quality, it seems that plant-based diets can exert a protective effect against NAFLD. 

#### 4.6.4. Dietary Approaches to Stop Hypertension (DASH Diet)

DASH is a sodium-restricted diet (<2400 mg/day) that is rich in fruits, vegetables, whole grains, low-fat dairy products, and lean protein; it is similar to the MedDiet but has a special focus on reducing total sodium intake [158]. Evidence shows that the DASH diet improves some risk factors for NAFLD (T2DM, MetS, obesity, dyslipidaemia). Furthermore, a systematic review and meta-analysis reported that DASH might be a better choice for weight management and reduction than other low-energy diets [159]. Some observational studies suggest that this diet could play a preventive role in NAFLD [160,161]. 

In fact, small short-term clinical trials have observed that NAFLD patients who adhere to a DASH diet achieve a reduction in body weight/BMI and improve triglyceride, ALT, AST, insulin sensitivity, and inflammatory marker serum levels [162]. Unfortunately, however, compliance with the diet remains low [163]. Therefore, it is thought that the DASH diet can be a useful tool to tackle NAFLD, but comparative clinical trials are still needed to assess the real effects of this diet on specific markers of NAFLD (histological and metabolic parameters). 

#### 4.6.5. Intermittent Fasting

Intermittent fasting (IF) is an emerging nutritional approach for weight loss and for the management of metabolic diseases and is considered an alternative to continuous calorie restriction [164]. The basic concept of IF involves regularly alternating periods of eating and fasting with either total food abstinence or very low energy intake [165]. IF has many forms, with the most common being alternate-day fasting (ADF), time-restricted feeding (TRF), or prolonged fasting. A recent systematic review and meta-analysis that included interventional trials concluded that IF was comparable to continuous energy restriction for short-term weight loss in individuals with overweight and obesity [166]. Since the primary goal of NAFLD therapy is weight loss, it is plausible that IF positively influences liver steatosis and metabolic markers. Another key mechanism responsible for the beneficial effects of IF (apart from the global caloric restriction) appears to be the body’s preferential shift from glycogenolysis-derived glucose to lipolysis-derived ketones [167]. Therefore, these protocols may have potential as a useful intervention in NAFLD patients. However, there is not yet scientific evidence to confirm these theoretical protective effects of IF against NAFLD.

## 5. Discussion

Current knowledge about NAFLD offers both certainties and unresolved doubts. On the one hand, we can be certain of the following: liver steatosis is the most common cause of chronic liver disease worldwide and a growing incidence is expected due to unhealthy lifestyle habits; nutrition is the main factor contributing to the development of NAFLD; adherence to a healthy diet, increased physical activity, and decreased body weight are the basis of the therapeutic approach to reduce the incidence of NAFLD and, in some circumstances, reverse hepatic steatosis; and general dietetic recommendations for these patients must include moderate weight loss (5%), maintenance of a low-calorie diet, cessation of alcohol and tobacco consumption, low carbohydrate intake (≤ 40% of daily energy intake), decreased fat (especially SFAs and TFAs), and simple carbohydrate consumption [69]. However, aside from these general certainties, many questions arise regarding the natural history, pathogenesis, and treatment of NAFLD. For example, are there specific foods or nutrients that can change the natural history of NAFLD? Which dietary pattern is best at dealing with this disease? Does diet composition play a role in the development of hepatic steatosis? Below, we describe the scientific evidence available to answer these questions. 

Weight reduction is a cornerstone in NAFLD management, and the goal should be to achieve ≥10% overall body weight loss. In fact, a lower reduction (3–5%) has been associated with decreased liver steatosis; however, it is necessary to reach a greater weight reduction (7–10%) in patients with NASH [67]. Having said this, weight reduction should not exceed 1.6 kg/week so as to avoid a worsening of fibrosis and hepatocyte necrosis [68]. Nevertheless, some authors have suggested that improving diet composition may reduce NAFLD even without changes in body weight [68].

High consumption of simple, high-glycaemic-index carbohydrates is associated with NAFLD [70]. Therefore, lower consumption of sugars should be advised, especially in high-calorie diets, although the pernicious effect of carbohydrates is more controversial in isocaloric or hypocaloric diets [89,90]. Fructose, as a paradigmatic example of this type of carbohydrate, deserves special attention. The intake of SSBs has increased by more than 40% in the last three decades, and this rise is one of the major underlying causes of chronic metabolic diseases, including NAFLD [73,74,75,76,77]. In fact, numerous experimental and observational studies have identified a clear relationship between high fructose consumption and NAFLD [78,79,80,81,82]. Moreover, it is not clearly known if there is a safety threshold for the consumption of sugary foods in the prevention of NAFLD [83]. It is noteworthy that the source of fructose (fruits or sugary drinks) might have different effects on the development of hepatic steatosis. Therefore, the intake of fructose from soft drinks is a risk factor for the development of NAFLD, while fructose from fruits is not. This could be explained by the lower fructose content in fruits and because they contain beneficial phytochemicals, micronutrients, and fibre, all of which provide prolonged satiety and a healthy GM [94].

Fat consumption has a dual effect on NAFLD development. On the one hand, MUFAs and specific types of PUFAs have a preventive effect against liver steatosis, whereas SFAs and TFAs have a negative effect [101,102,105,106,123]. Based on animal and human studies, it has been proposed that fats induce liver steatosis through several mechanisms, including IR, endotoxemia, dysbiosis, inflammation, and increased multiple plasma ceramides. Although there is consensus that excessive fat intake has a deleterious effect on the liver, most of the data have been drawn from observational studies, and only a limited number of small clinical trials on this issue have been published. On the other hand, MUFAs consumption seems to exert a protective effect against NAFLD, as observed in a sub-cohort study of the PREDIMED trial [119]. 

There are few data related to the effect of protein consumption on the development of NAFLD, and the results obtained are quite heterogeneous. It has been reported that high protein intake could have a negative, neutral, or even positive effect on NAFLD [125,126,127,128]. The best explanation for this controversy is that animal and vegetable proteins could exert opposite effects. Thus, high animal protein intake could have a negative effect, whereas vegetable protein might have a beneficial effect on NAFLD [129]. However, the supposed problem with animal proteins may be influenced by bias since they are usually consumed jointly with SFAs (meat products). 

It has been proposed that the quality and composition of carbohydrates and fat intake may be more relevant than their specific amounts on an individual’s total caloric intake. For instance, in a randomised study with isocaloric diets, it was observed that fatty liver decreased by 20% and increased by 35% in patients on low- or high-fat diets, respectively. These changes were not related to weight loss, suggesting that the macronutrient composition of the diet is as important as weight loss to prevent or treat NAFLD. However, the best dietary composition for NAFLD treatment, independent of weight loss, remains to be determined [68].

Based on the scientific evidence available, the best dietary pattern to prevent or treat NAFLD should promote the exclusion of sugary foods and SSBs, along with the reduction of red and processed meats and animal fat, and enhance the consumption of complex carbohydrates, vegetables, fruits, legumes, whole grains, nuts, fish, seeds, and olive oil, which is abundant in MUFAs and Omega-3 PUFAs [82]. These recommendations are present in several dietary patterns, such as the MedDiet, the vegetarian/vegan diet, and the DASH diet. Nonetheless, it has been reported that long-term adherence to these diets was lower in the case of the DASH diet, and higher in the vegetarian/vegan diet and, especially, in the MedDiet. Indeed, the MedDiet was proposed in the most recent guidelines as the best dietary pattern to prevent or treat NAFLD and CVD [14,153]. Besides dietary intervention, some reports suggest that a multifactorial intervention including moderate caloric restriction, increased physical activity, and weekly meetings to reinforce lifestyle changes is more effective to protect against NAFLD, compared to a dietetic intervention alone [68].

This review highlights the lack of high-quality randomised controlled trials to determine the best evidence-based approach to NAFLD prevention and treatment, including nutrients, dietary composition, or dietary patterns. Therefore, further nutritional intervention trials are needed to evaluate the beneficial effect of nutrition in NAFLD and related diseases, such as the PREDIMED study which demonstrated the efficacy of the MedDiet in the reduction of CVD incidence [69].

## 6. Conclusions

Current evidence has shown the negative influence of hypercaloric diets, excessive consumption of fats (mainly SFAs, TFAs, and Omega-6 PUFAs), and intake of added sugars (mainly fructose) in the pathogenesis of NAFLD. Currently, nonpharmacologic measures (healthy diet and physical activity) are the only effective approach to treat or prevent NAFLD. Therefore, we should propose to our NAFLD patients or those at high risk for its development to avoid/decrease these negative factors and to follow a health-promoting diet such as the MedDiet or other diets with similar features (vegetarian or DASH) to achieve a significant weight reduction and modify the natural history of hepatic steatosis [69,149].

## Figures and Tables

**Figure 1 nutrients-13-01442-f001:**
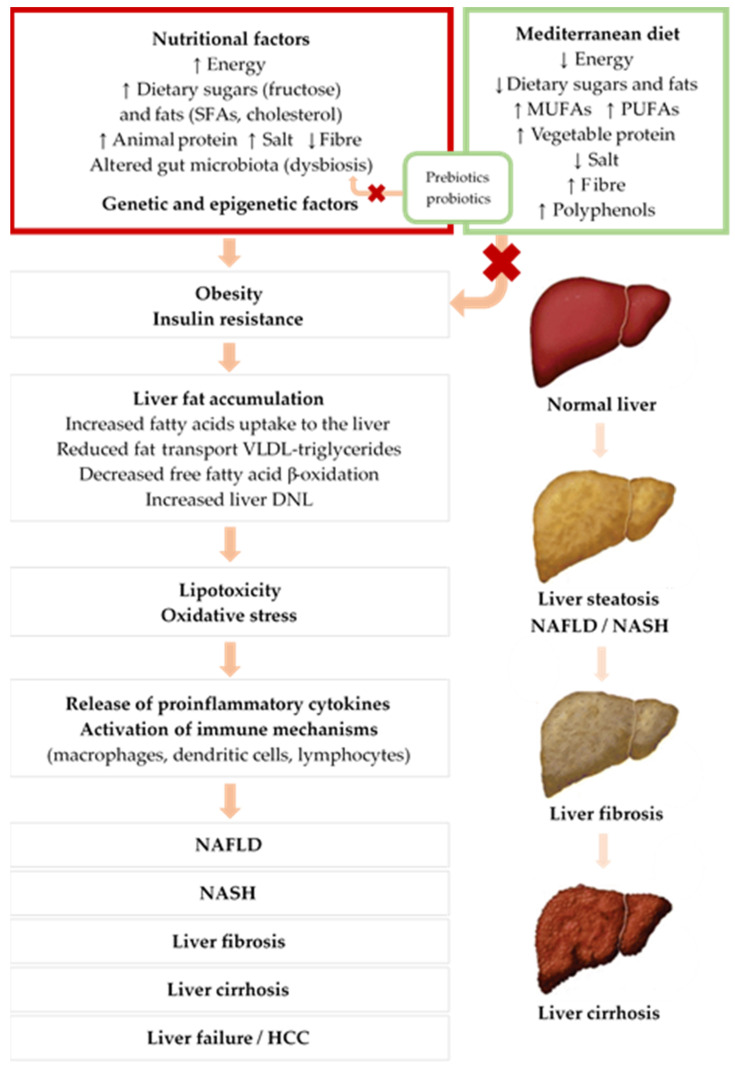
Mechanisms of NAFLD development and the influence of nutritional factors. Upwards pointing arrows (↑) indicate increased intake, downwards pointing arrows (↓) indicate decreased intake. Abbreviations: SFAs: saturated fatty acid; MUFAs: monounsaturated fatty acids; PUFAs: polyunsaturated fatty acids; VLDL: very low-density lipoprotein; DNL: de novo lipogenesis; NAFLD: non-alcoholic fatty liver disease; NASH: non-alcoholic steatohepatitis; HCC: hepatocellular carcinoma.

**Figure 2 nutrients-13-01442-f002:**
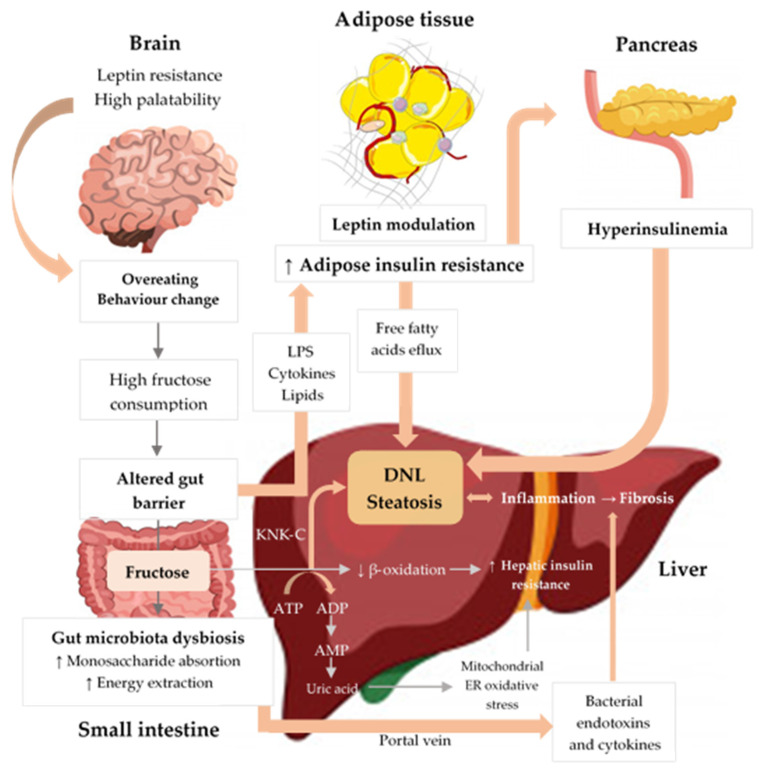
Excessive fructose intake is associated with inflammation, cellular stress, dysbiosis, and hepatic steatosis. Fructose strongly activates DNL, inducing IR and liver inflammation. Fructose metabolism by fructokinase (KHK–C) leads to uric acid generation, mitochondrial dysfunction, and oxidative stress, which contributes to hepatic IR and inflammation. Additionally, high fructose consumption induces gut microbiota dysbiosis, which increases gut permeability, leading to translocation of bacterial endotoxins, cytokines, and lipopolysaccharide (LPS), thus driving hepatic inflammation and IR. Finally, fructose increases peripheral IR through LPS, cytokines, and lipid oxidation and negatively influences appetite through gut–brain axis alterations, high palatability, and leptin modulation, thereby promoting increased energy intake and weight gain [95,96,97,98,99]. Upwards pointing arrows (↑) indicate increase, downwards pointing arrows (↓) indicate decrease.

**Table 1 nutrients-13-01442-t001:** Epidemiological and pathological differences between NAFLD and NASH, and their respective minimum weight loss goals in dietary treatment [13,14].

	NAFLD	NASH
Overall prevalence (estimated in general population)	25%	3–5% (7–30% of NAFLD patients)
Pathology	(a) Steatosis alone (b) Steatosis with lobular or portal inflammation, without ballooning (c) Steatosis with ballooning but without inflammation	Steatosis, ballooning, and lobular inflammation
Fibrosis progression	1 stage of progression over 14 years	1 stage of progression over 7 years
Liver cirrhosis 10–20-year time	2–3%	15–20%
HCC incidence rate	0.44 per 1000 person years	5.29 per 1000 person years
Liver-specific mortality incidence rate	0.77 per 1000 person years	11.77 per 1000 person years
Overall mortality incidence rate	15.44 per 1000 person years	25.56 per 1000 person years
Pharmacological treatment	Non approved	Non approved
Weight loss goal in dietary treatment	3–5% of weight loss	7–10% of weight loss

Abbreviations: NAFLD: non-alcoholic fatty liver disease; NASH: non-alcoholic steatohepatitis; HCC: hepatocellular carcinoma.

**Table 2 nutrients-13-01442-t002:** Non-invasive biomarker detection methods [53,56].

Steatosis and Fibrosis Scores	Measured parameters
Fatty liver index (FLI)	BMI, waist circumference, triglycerides, and GGT
NAFLD liver fat score (NLFS)	Metabolic syndrome, T2DM, fasting serum insulin, and fasting serum AST/ALT ratio
NAFLD fibrosis score (NFS)	Age, BMI, IFG and diabetes, AST-to-ALT ratio, platelets, and albumin
Fibrosis-4 (FIB-4) index	Age, AST, ALT, and platelet

Abbreviations: BMI: body mass index; GGT: gamma-glutamyl transferase; T2DM: type 2 diabetes mellitus; AST: aspartate aminotransferase; ALT: alanine aminotransferase; IFG: impaired fasting glucose.

**Table 3 nutrients-13-01442-t003:** Comorbidities associated with NAFLD and NASH, respectively, based on a meta-analysis [13].

Comorbidities	NAFLD	NASH
Obesity	51%	82%
Hypertension	39%	67%
Type 2 diabetes mellitus	23%	47%
Hyperlipidaemia/dyslipidaemia	69%	72%
Metabolic syndrome	41%	71%

Abbreviations: NAFLD: non-alcoholic fatty liver disease; NASH: non-alcoholic steatohepatitis.

## Data Availability

Data sharing not applicable.

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
