# Peer review of "Overview of Non-Alcoholic Fatty Liver Disease (NAFLD) and the Role of Sugary Food Consumption and Other Dietary Components in Its Development"

_nutrients, 2021, doi:10.3390/nu13051442_

Round 1

Reviewer 1 Report

The authors have prepared a narrative review that describes NAFD and the role of sugary food consumption and other dietary components in its development. While title is focused on diet and its components, the manuscript objectives are to provide an overview of the general aspects of NAFLD and diet. I think these two objectives, are reasonably complex individually, and attempting to provide a comprehensive review on both these topics detracts from the manuscript quality, and at times is confusing to the reader. The authors should declare one area of focus and develop it more critically. For example, the diet story is more complex than advertised. The gut microbiome is an important mechanism that can drive NAFLD. However, this is not mentioned in the disease pathophysiology, until the very end. In addition, dietary fiber, and plant based diets are important determinants of the microbiome and these can induce changes very rapidly. A critical discussion of dietary fiber is needed. The fatty intake and NAFLd section is disjointed. There is discussion in both animal and humans, but it is not organized, confusing the reader. The epidemiology section discusses omega-6 fats in detail, but the omega 3 discussion is limited a broad statement about systematic reviews without details. There is no discussion about supplementation compared to whole foods. Regarding the section about protein intake and NAFLD, the authors discuss studies about protein supplementation and NAFLD outcomes. However, there is limited discussion on other dietary components, and overall energy consumption during the intervention period. These details are need to help the reader understand whether the benefit is a consequence of protein, or energy modulation. “In general, dietary intervention is more effective in normalizing aminotransferases and lowering intrahepatic fat, whereas exercise improves insulin sensitivity and decreases BMI [44]” This is not clear. There is no comment here about the role of weight loss. Do the authors mean to say that dietary intervention that leads to weight loss is effective to normalize liver enzymes, or that diet can achieve this independent of weight loss? Did the authors develop figure 1 or was this adapted from another source? It would be interesting for the authors to include another figure that outlines the mechanisms for NAFLD development, and identify in the figure the various places through which diet (broadly) may influence the pathophysiology. Integrating pathophysiology strategically with diet mechanisms could provide a unique angle to differentiate this review from others. “NASH can be classified from mild (F0–F1) to advanced fibrosis (≥F3, bridging) and 74 finally cirrhosis (F4).” This statement refers to fibrosis severity, but does not define NASH. Please describe the NASH lesion more accurately with references. The epidemiology of NAFLD would benefit from a section about “lean NAFLD” “Among the proposed 124 hit factors, many of them can interact with each other, forming a vicious circle” – Try and retain scientific writing language. Pls consider alternate terms to replace “vicious circle” “2016 NALFD” – There is a typo here Is Table 1 taken from reference 33? If so, I don’t think it is necessary to replicate the table in this text. It doesn’t really add much, and it is not a central topic of the manuscript. Please consider deleting this and summarizing it in text. Please include reference for Transient Elastrography and Fibroscan. . “The fibrosis progression rate is estimated to be 1 stage every 14 263 years (NAFLD) or 7 years (NASH); the rate is doubled by arterial hypertension” This statement is not clear. Please revise for clarity. What is the significance of this? “NAFLD patients should be forced to maintain a low-caloric diet and 289 stop alcohol and tobacco consumption” Authors need to be careful about their use of language. “forced” is not an appropriate term. “Encouraged” or “supported” may be better placed.

Reviewer 2 Report

line# 29- etopic is inappropriate.  (would change to excessive)

line# 68- smaller subgroup. (would be better to have a % instead)

line# 69- would advise to add hepatic stellate cells activation, change scar tissue to fibrous  tissue.

line# 70-71,  consider replacing fibroblast by "fibrillar collagen and other extracellular matrix protein" which compromise hepatic function and structure. 

line#72 change fibroblasts to fibrosis.

line# 101 prevalence of NAFLD in Asia varies to higher percentage needs to be cited.

line# 285 Any drug to be changed to "no standardized or specific approved medications in treating NAFLD"

line# 289 change "forced" to "advised"

line# 540 EVOO spells out as extra virgin olive oil.

line# 624 and line # 649 Clarify what Bacteroidetes concentration is harmful and causing NAFLD

line# 703 sentence unclear 

Reviewer 3 Report

The review is an good addition to the field of liver related diseases specifically in the field of NAFLD. There are few minor questions I have for this review: 

  1. In the method section, the authors nicely described what type of studies were included in the review. These are specifically related to mouse models. However, what type of study was not included and a brief reasoning justifying such would be helpful.
  2. Page 3, lines 103-104 - it is mentioned with reference that western diets influencing epidemiology of the disease. This sentence needs to further clarified as to how this happened by not only citing papers on this topic but very briefly describing them as well. 
  3. Overall it is a nicely written article on NAFLD. However at the beginning of the article, the authors compares the definition with NASH. Therefore a short paragraph about the the two disease comparison would be helpful. Alternatively a table comparing the two with main key points (such as epidemiology, pathology, effect of different diets, treatment) would be helpful to understand and differentiate this disease from NASH with a deeper understandings. 

Round 2

Reviewer 1 Report

The authors have made substantial revisions that have enhanced the manuscript. Figure 1 should include more details prior to acceptance. These could include adding in the role of microbiome, and where it fits in the pathogenesis, and how prebiotics, including fiber, and probiotics may be used for effect. Major edit for English flow and grammar is required.

Author Response

Dear Reviewer,

We have introduced in the last version (v25 Nutrients) of the manuscript your suggestions:

  • Figure 1 has modified to include new information
  • We have done a thorough revision of English
  • We have detected and corrected some errors in the bibliographic references

Sincerely yours, 

E. Sacanella